# Academics–Athletics Conflict and College Athletes’ Well-Being: The Mediating Effect of Negative Emotions and the Moderating Effect of Life Motivation

**DOI:** 10.3390/bs13020093

**Published:** 2023-01-22

**Authors:** Wujun Sun, Lei Liu, Yuan Jiang, Ping Fang, Xiaosheng Ding, Guangjun Wang

**Affiliations:** 1Faculty of Education, Henan Normal University, Xinxiang 453007, China; 2School of Labor and Human Resources, Renmin University of China, Beijing 100872, China; 3Department of Psychology, Beijing Sport University, Beijing 100084, China; 4Department of Psychology, Capital Normal University, Beijing 100048, China; 5Department of Physical Education, Beijing Union University, Beijing 100101, China

**Keywords:** academics–athletics conflict, life satisfaction, negative emotions, eudaimonic motives, hedonic motives

## Abstract

For college athletes who perform dual roles (student and athlete), the academics–athletics conflict is inevitable in daily life. Although existing studies have focused on the adverse effects of this conflict on the well-being of college athletes, they have not yet determined the underlying mechanism and effective measures to alleviate it. To explore the underlying mechanism of academics–athletics conflict, which affects the well-being of college athletes, we constructed a moderated mediating model to examine the critical role of negative emotions and life motivation in the relationship between them. The study randomly selected 802 college athletes from China to examine the relationships between academics–athletics conflict, negative emotions, eudaimonic motives, hedonic motives, and life satisfaction. The results showed that (1) negative emotions played an important mediating role between academics–athletics conflict and college athletes’ life satisfaction, with more than 79% of the effect of academics–athletics conflict being achieved through negative emotions. (2) Eudaimonic motives significantly moderated the first half of the mediation path of negative emotions between academics–athletics conflict and life satisfaction. Individuals with high eudaimonic motives experienced fewer negative emotions in the medium and weak conflict conditions. (3) Hedonic motives had a significant moderating effect on the second half of the mediation path. Individuals with high hedonic motives had greater life satisfaction across negative emotion conditions. This study provides important insights for a comprehensive understanding and in-depth study of the relationship between conflict and the well-being of college athletes, as well as a reference for the quality-of-life enhancement and motivation development for college athletes.

## 1. Introduction

Self-determination theory suggests that setting goals and striving to achieve them is an important part of personal development [1,2]. In this process, the simultaneous pursuit of multiple meaningful goals and conflicts between goals often occur [3,4]. This is especially true for college athletes who play the dual role of student and athlete. They need to work hard to achieve and maintain a certain level of academic performance, while also maintaining optimal athletic performance through intense physical training to compete [5]. Conflicts between academic and athletic goals often exist due to resource constraints such as time and energy. Existing studies have shown that academics–athletics conflicts cause college athletes to experience more psychological stress and emotional problems than the average students, which has potentially negative effects on their physical and mental health and well-being [6,7,8]. Although these studies have focused on the adverse effects of academics–athletics conflict, they have not been able to identify the underlying mechanisms and effective measures to mitigate it. Therefore, exploring the issue of the influence mechanism of academics–athletics conflict on college athletes’ well-being has positive implications for a comprehensive understanding of how goal conflict affects the quality of life of college athletes and how to prevent and ameliorate its negative effects.

### 1.1. Academics–Athletics Conflict and College Athletes’ Well-Being

As a specific manifestation of goal conflict in college athletes, there seems to be no doubt about the negative relationship between academics–athletics conflict and well-being. Theoretical studies of multi-goal pursuit suggest that conflict is often accompanied by inhibition of goal behavior and obstruction of goal progress [4,9,10]. This conflict triggers an increase in individual psychological stress and a decrease in the experience of well-being [11,12]. Scholars focusing on college athletes have also found that academics–athletics conflict significantly negatively impacts students’ school satisfaction, well-being, or psychological well-being [6,8,13]. Unfortunately, however, recent research has questioned this seemingly strong negative relationship. Research has found that the negative effect of goal conflict on well-being is not unchanged, and the relationship between the two may vary depending on the individual or situational factors [14]. The realities of college athletes also suggest that even when everyone faces academics–athletics conflict, some students can still maintain good academic and athletic performance and excellence [15]. There may also be important mediating or moderating variables between academics–athletics conflict and college athletes’ well-being, causing the relationship between the two to show different results in different situations or conditions. This speculation provides an opportunity to prevent and ameliorate the negative effects of academics–athletics conflict. It is necessary to explore this problem in depth and analyze the boundary conditions and paths of academics–athletics conflict affecting well-being to deepen understanding of the relationship between them. Life satisfaction is an important index to measure life quality and well-being, which is widely used in the study of the relationship between goal conflict and well-being [16]. This study also uses life satisfaction to reflect college athletes’ evaluation of their experience of happiness. Based on the review of previous theoretical and empirical studies, Hypothesis 1 is proposed:

**Hypothesis** **1.**
*Academics–athletics conflict negatively impacts the college athletes’ life satisfaction.*


### 1.2. The Role of Negative Emotions in Academics–athletics Conflict and College Athletes’ Well-Being

Emotion connects life experience with goals and actions and is integral to exploring the relationship between academics–athletics conflict and well-being [14,17]. Many studies on goal conflict have confirmed the negative valence characteristics of incompatible goals in inducing individual emotional experiences. Control Theory states that conflicts within the goal hierarchy can lead to distress [18]. Individuals experiencing goal conflicts tend to ruminate, hesitate, and report greater negative emotions, including depression, frustration, and mental anxiety [11,19]. Studies of college athletes have also shown that the goal conflicts between academic work and athletic exercise bring negative emotions such as shame and frustration to college students and that academics–athletics conflict is significantly and positively correlated with negative emotional experience [13,17]. Positive emotional experiences enhance individuals’ perception and evaluation of well-being and increase life satisfaction, while negative emotions do the opposite [20]. Adverse emotional states such as depression, anxiety, and frustration induced by academics–athletics conflict will permeate individuals’ learning and exercise, perhaps becoming a critical factor in impairing their life satisfaction, health, and well-being [13]. Accordingly, this study proposes Hypothesis 2:

**Hypothesis** **2.**
*Negative emotions mediate the relationship between academics–athletics conflict and college athletes’ life satisfaction. Academics–athletics conflict decreases college athletes’ life satisfaction by increasing their negative emotional experience.*


### 1.3. The Role of Life Motivation

Eudaimonic and hedonic motives are two different sources of motivation for individuals to pursue a “good life” [21]. These two life motives reflect the different attitudes and principles that people follow in their daily lives. Eudaimonic motives are based on the principle of achieving one’s self-worth and potential and affirm the value of struggle. Hedonic motives are based on the principle of pleasure and comfort and advocate that people should enjoy themselves in time [22]. Life motivation has often been studied as an essential factor influencing well-being. These studies suggest that the different value orientations contained in the two life motivations may cause individuals to have different attitudes, emotions, and behavioral responses to the situation they are in, which in turn determines how much happiness they experience in their goal pursuit [21,23]. Moreover, recent studies on emotion regulation further suggest that eudaimonic and hedonic motives are essential factors that drive individuals to regulate their emotions [24]. Although only mildly associated with increases in positive emotions, both motivations significantly decrease negative emotions, with eudaimonic motives contributing more to emotional regulation and well-being than hedonic ones [25]. Based on the close relationship between eudaimonic and hedonic motives and emotional regulation and well-being, the study proposes Hypothesis 3:

**Hypothesis** **3.**
*Eudaimonic and hedonic motives moderate the mediating path of academics–athletics conflict affecting college athletes’ life satisfaction through negative emotions. The three possibilities included in this hypothesis are shown in (a), (b), and (c) in Figure 1, respectively. (a) Eudaimonic and hedonic motives simultaneously moderate the first and second halves of the mediation path. (b) Eudaimonic motives moderate the first half of the mediation path, while hedonic motives moderate the second half. (c) Hedonic motives moderate the first half of the mediation path, while eudaimonic motives moderate the second half.*


## 2. Methods

### 2.1. Participants

The study recruited participants and collected data across China using the Wenjuanxing platform, https://www.wjx.cn/ (accessed on 17 December 2022). The 802 college athletes in the study were from China. All participants were between the ages of 16 and 25 years old, with an average age of 20.01 (*SD* = 1.44). The study was comprised of 450 males (56.1%) and 352 females (43.9%). With regard to class rank, the study included 239 freshmen (29.8%), 267 sophomores (33.3%), 184 juniors (22.9%), and 112 seniors (14.0%). Furthermore, 395 (49.3%) participants majored in social sciences, whereas 407 (50.7%) participants majored in natural sciences. All participants provided informed consent prior to participation in the study.

### 2.2. Measurements

#### 2.2.1. Conflicting Goals Scale

The Conflicting Goals Scale, adapted by Berrios et al. based on the Strivings Instrumentality Matrix, was used [26]. The scale asked participants to answer three questions to assess the extent to which these goals have conflicted over the past few days, based on the individual’s five most important current goals. In this study, to obtain the degree of conflict between academic and athletic goals, the participating college athletes were asked to recall three academic and three athletic goals they worked on in the past two weeks and answer all questions based on the relationship between the two types of goals. The scale uses a 5-point Likert scale ranging from 1 (strongly disagree) to 5 (strongly agree), with higher scores indicating higher levels of goal conflict. This method is less time-consuming and more convenient than the original scale [14]. In this study, Cronbach’s alpha coefficient of the Conflicting Goals Scale is 0.748, indicating good internal consistency.

#### 2.2.2. Positive Affect and Negative Affect Scale

The negative emotions subscale of the Positive Affect and Negative Affect Scale developed by Bradburn was used to assess the frequency of negative emotions experienced by participants in the last two weeks [27]. The subscale consists of six items on a 4-point Likert scale ranging from 1 “not at all” to 4 “often”, with higher scores indicating a higher frequency of negative emotions. In this study, the fit indices of the negative emotions’ subscale were *x*^2^/*df* = 3.122, RMSEA = 0.051, GFI = 0.989, NFI = 0.984, IFI = 0.989, TLI = 0.980, CFI = 0.989, and the Cronbach’s alpha coefficient was 0.829.

#### 2.2.3. Hedonic and Eudaimonic Motives for Activities Questionnaire

The Hedonic and Eudaimonic Motives for Activities Questionnaire developed by Huta and Ryan was used to evaluate the life motivation of participants [22]. The scale consists of two subscales: eudaimonic motives and hedonic motives. These two subscales are used to assess the extent to which people engage in activities based on eudaimonic and hedonic principles in their daily lives, respectively. The scale includes nine items on a 7-point Likert scale ranging from 1 “not at all” to 7 “very much”, with higher scores on each subscale indicating a higher tendency toward eudaimonic or hedonic behavior. In this study, the fit indices of the scale were *x*^2^/*df* = 3.309, RMSEA = 0.054, GFI = 0.981, NFI = 0.982, IFI = 0.988, TLI = 0.980, CFI = 0.987. The Cronbach’s alpha coefficients of eudaimonic and hedonic motives subscales were 0.893 and 0.865, respectively.

#### 2.2.4. Satisfaction with Life Scale

The Satisfaction with Life Scale, created by Diener et al., was used to evaluate the overall satisfaction of college athletes with their personal lives [28]. The scale consists of five items and is rated on a 7-point Likert scale from 1 “strongly disagree” to 7 “strongly agree”, with higher scores indicating higher levels of life satisfaction. The scale is one of the most widely used instruments to measure life satisfaction and has good reliability and validity [14]. In this study, the fit indices of life satisfaction scale were *x*^2^/*df* = 3.361, RMSEA = 0.054, GFI = 0.995, NFI = 0.996, IFI = 0.997, TLI = 0.990, CFI = 0.997. The Cronbach’s alpha coefficient of the scale was 0.888.

### 2.3. Statistical Analysis

SPSS 21.0 statistical software was used to conduct common method bias test, descriptive statistics and calculate correlation coefficients of key variables. The proposed moderated mediating effects model was tested using PROCESS macro 2.16 for SPSS according to the test proposed by Hayes [29]. The test was conducted in two steps. First, we use Model 4 to test the mediating effect of negative emotions between academics–athletics conflict and life satisfaction. We then used Models 58 and 21 to test the moderating effect of eudaimonic and hedonic motives in the two halves of the mediating path. In addition, we chose the bias-corrected percentile bootstrap method for testing mediating effects. Standard errors and confidence intervals for the parameter estimates were obtained from 5000 bootstrap samples (802 participants per sample). Finally, we used a simple slope test to determine how eudaimonic and hedonic motives moderated the relationship between academics–athletics conflict, negative emotions, and life satisfaction.

## 3. Results

### 3.1. Common Method Biases Test

The study used Harman’s one-way test for common method bias for all items of the four scales. The results showed that five factors with eigenvalues greater than one were extracted. The amount e of variance explained by the largest common factor obtained without rotation and after rotation was 26.09% and 15.34%, respectively, which were less than the critical criterion of 40% [30]. Therefore, this study was subject to a very low degree of common method bias.

### 3.2. Descriptive Statistics and Correlational Analysis

The results of descriptive statistics and correlational analysis for all variables are shown in Table 1. The academics–athletics conflict was significantly and positively correlated with negative emotions and hedonic motives, significantly and negatively correlated with life satisfaction, and insignificantly correlated with eudaimonic motives. Negative emotions were significantly and negatively correlated with eudaimonic motives and life satisfaction, and insignificantly correlated with hedonic motives. Eudaimonic motives were significantly and positively correlated with hedonic motives and life satisfaction. Hedonic motives were significantly and positively related to life satisfaction. Gender was significantly and negatively related to life satisfaction.

### 3.3. Mediating Model Analyses

Correlation analysis showed that the relationship between academics–athletics conflict, negative emotions, and life satisfaction meets the conditions of mediating effect test. Given the significant correlation between gender and life satisfaction, this study included gender as a control variable in the model. Results (see Table 2) showed that academics–athletics conflict significantly and positively affected negative emotions (*t* = 8.51, *p* < 0.001). When negative emotions were added to the model as a mediating variable, it had a significant negative impact on life satisfaction (*t* = −5.13, *p* < 0.001). Academics–athletics conflict, although negatively affecting life satisfaction, was not significant (*t* = −0.37, *p* > 0.5). The control variable of gender significantly and negatively affected life satisfaction (*t* = −2.73, *p* < 0.01). The 95% confidence interval of indirect effect was [−0.08, −0.03], indicating that negative emotion significantly mediated between academics–athletics conflict and life satisfaction (as shown in Figure 2), accounting for 79.77% of the total effect.)

### 3.4. Moderated Mediating Model Analyses

Hypotheses 2(a) were tested using Model 58 of SPSS’s PROCESS macro program and Hypotheses 2(b) and 2(c) were tested using Model 21. The results showed that only Hypothesis 2(b) was verified, as shown in Table 3. The direct effect of academics–athletics conflict on life satisfaction was not significant (*β* = 0.05, *t* = −1.53, *p* > 0.5). In the indirect path, academics–athletics conflict had a significant positive effect on negative emotions (*β* = 0.28, *t* = 8.26, *p* < 0.001), and eudaimonic motives significantly and negatively affected negative emotions (*β* = −0.10, *t* = −2.88, *p* < 0.01). Eudaimonic motives and academics–athletics conflict had significant interaction effects on negative emotions (*β* = 0.07, *t* = 2.28, *p* < 0.05). Negative emotions had a significant negative effect on life satisfaction (*β* = −0.20, *t* = −5.76, *p* < 0.001), hedonic motives significantly and positively affected life satisfaction (*β* = 0.30, *t* = 9.11, *p* < 0.001). Hedonic motives and negative emotions had significant interaction effects on life satisfaction (*β* = 0.06, *t* = 2.14, *p* < 0.05). The control variable gender had a significant effect on life satisfaction (*β* = −0.26, *t* = −3.87, *p* < 0.001). It is evident that eudaimonic and hedonic motives moderate the first and second half of the mediating effect of negative emotions, respectively.

The effect values and 95% confidence intervals for the moderated mediating effects are shown in Table 4. The moderating effect of eudaimonic motives and hedonic motives was analyzed at three levels: mean, mean plus one standard deviation, and mean minus one standard deviation). When the eudaimonic motives were on −1, negative emotions’ mediating effect sizes at the three levels of hedonic motives were −0.054, −0.042, and −0.030, respectively. Bootstrap 95% confidence interval were [−0.101, −0.022], [−0.078, −0.017], and [−0.074, −0.007] respectively. When eudaimonic motives were on 0, the mediating effect sizes of negative emotions at the three levels of hedonic motives were −0.072, −0.055, and −0.039, respectively. Bootstrap 95% confidence interval were [−0.115, −0.038], [−0.083, −0.033], and [−0.076, −0.012] respectively. When the eudaimonic motives were on +1, the mediating effect sizes of negative emotions at the three levels of hedonic motives were −0.089, −0.069, and −0.049, respectively. Bootstrap 95% confidence interval were [0.148, 0.047], [0.102, 0.042], and [0.088, 0.015] respectively.

### 3.5. Simple Slope Test

A simple slope test was further used to analyze the moderating effect of eudaimonic motives on the relationship between academics–athletics conflict and negative emotions in the first half of the mediating effect and the moderating effect of hedonic motives on the relationship between negative emotions and life satisfaction in the second half of the mediating effect.

The results of the moderating effect of eudaimonic motives are shown in Figure 3. At the high (M + SD), medium (M) and low (M − SD) levels of eudaimonic motives, the positive effect of academics–athletics conflict on negative emotions was significant (*simple slope*
_high_ = 0.35, t = 8.05, *p* < 0.001; *simple slope*
_medium_ = 0.28, t = 8.26, *p* < 0.001; *simple slope*
_low_ = 0.21, t = 4.55, *p* < 0.001). The effect of academics–athletics conflict on negative emotions was stronger in the condition of high eudaimonic motives. The moderating effect of eudaimonic motives was more pronounced when academics–athletics conflict was below the medium level (M or M − SD).

The results of the moderating effect of hedonic motives are shown in Figure 4. At high (M + SD), medium (M) and low (M − SD) levels of hedonic motives, negative emotions had a significant negative effect on life satisfaction (*simple slope* _high_ = −0.16, t = −3.89, *p* < 0.001; *simple slope*
_medium_ = −0.21, t = −6.343, *p* < 0.001; *simple slope*
_low_ = −0.27, t = −5.92, *p* < 0.001). Negative emotions had a stronger impact on life satisfaction in the condition of low hedonic motives. The moderating effect of hedonic motives was more pronounced when negative emotions were above the medium level (M or M + SD).

## 4. Discussion

### 4.1. Academics–Athletics Conflict and College Athletes’ Life Satisfaction

The results of this study indicated a significant negative association between academics–athletics conflict and college athletes’ life satisfaction. The findings are consistent with previous theoretical perspectives and empirical findings [6,13]. Control Theory and more recent motivation and personality theories argue that the emergence of goal conflict activates an individual’s behavioral inhibition system to check for threats and evaluate risks [9,18]. The activation of this system often leads to goal obstruction and negative evaluations, which are important factors in reducing individual life satisfaction [11,12]. Thus, academics–athletics conflict is often associated with distress or low life satisfaction.

However, it is worth noting that the negative effect of academics–athletics conflict on college athletes’ life satisfaction is unstable. In this study, academics–athletics conflict was significantly and negatively related to life satisfaction without considering negative emotions and life motivation, with the former significantly affecting the latter. Surprisingly, when examining the mediating effect of negative emotions and the moderating effect of life motivation, the path coefficient between academics–athletics conflict and life satisfaction became less significant (see Figure 2 and Table 3). This result verifies the findings of related studies in the field of goal conflict [14,31], indicating that there are indeed some important mediating or moderating variables between academics–athletics conflict and college athletes’ well-being, which need to be further investigated.

### 4.2. Mediating Effect of Negative Emotions

Emotion is essential in shaping one’s life satisfaction judgments and plays a vital role in the influence of individual and environmental factors on happiness [32,33]. This present study suggests that the effect of academics–athletics conflict on college athletes’ life satisfaction is achieved mainly through negative emotions. The relationships between the variables involved in the results are consistent with previous theoretical and empirical findings [11,13,17] and verify the findings of the mediating effect of emotions [21,33,34]. According to the Control Theory, academics–athletics conflict does bring varying degrees of distress and negative emotions to college athletes and become an important factor affecting their negative evaluation of life status [13,18].

The results of the mediating effect test also prove the importance of changing negative emotions. More than 79% of the effects of academics–athletics conflict are mediated through negative emotions. These results suggest that we can change any segment of this mediation path, which can largely reduce the adverse effects of academics–athletics conflict on life satisfaction. For example, we can guide students to group conflicting academic and athletic goals into a more general long-term goal (such as being an achiever), increasing the degree of commonality between goals and goal self-concordance [35]. By doing this, we can change individuals’ negative emotional experiences to some extent and, in turn, increase their life satisfaction. We can also reduce the adverse impact of negative emotions on life satisfaction by improving individual psychological resilience [36] or increasing psychological resources [6,37]. Of course, from the perspective of emotional regulation, there are also important ways to reduce negative emotions and their negative effects, such as developing motivation that contributes to emotion regulation [25], choosing appropriate emotion regulation strategies [38] and increasing self-efficacy for emotion regulation [39].

### 4.3. Moderating Effect of Eudaimonic Motives and Hedonic Motives

This study found that eudaimonic motives had a significant moderating effect on negative emotions in the first half of the mediation path between academics–athletics conflict and life satisfaction. Individuals high in eudaimonic motives experienced less negative emotions in the medium and low conflict conditions. Hedonic motives had a significant moderating effect on the second half of the mediation path. Individuals with high hedonic motives had higher life satisfaction in all negative emotional conditions. This result verified the previous findings on the moderating effects of eudaimonic and hedonic motives on emotion generation and experience [24,25]. It also added details to this moderating effect, showing the differences between the two motivations in regulating negative emotions and their negative effects. According to the Levels of Valence Model theory, an individual’s overall emotional experience is derived from multidimensional evaluations at the micro level, such as goal conduciveness, self-congruence, and moral goodness [40]. Eudaimonic motives, which emphasize the development of one’s potential and self-worth, make it easier for individuals to see the potential value and meaning of goal conflict as a necessary path to achieve their ultimate goals. As a result, they can reduce the negative valence in the emotional evaluation and decrease individuals’ negative emotional experiences [21,23]. Therefore, the moderating effect of eudaimonic motives on emotions may occur more during the generation of negative emotions through altered cognitive appraisals. Hedonic motives, which are oriented towards pleasure, enjoyment, and comfort, may be more sensitive to the presence of negative emotions. Once individuals perceive negative emotions, hedonic motives may alleviate negative emotions by avoiding negative events (e.g., academics–athletics conflict) and seeking out temporary positive events (e.g., traveling and playing games) [22]. Therefore, the moderating effect of hedonic motives on emotions is more likely to occur after negative emotions are generated and achieved through avoidance or transfer.

In real life, it is not an either-or relationship for individuals to have eudaimonic and hedonic motives, people can pursue both life motivations simultaneously. Depending on how individuals perform on both motivations, their lifestyles can be categorized into four types: full life (high eudaimonic and high hedonic), eudaimonic life (high eudaimonic and low hedonic), hedonic life (low eudaimonic and high hedonic), and empty life (low eudaimonic and low hedonic) [22,23]. According to this study’s results, when both eudaimonic and hedonic motives are higher, individuals are likely to experience fewer negative emotions, and their life satisfaction is likely to be higher. This finding is consistent with the findings of previous studies that a full lifestyle brings more happiness [22,41]. Meanwhile, it also provides a valuable reference for improving college athletes’ quality of life and happiness through developing good life motivation. From a long-term and developmental perspective, educators can guide students to form the proper value orientation in life so that they know how to not only accept the present moment and enjoy the happiness of life but also look to the future and keep striving to surpass themselves. From a contextual adaptation perspective, when students encounter academics–athletics conflicts, they can also use eudaimonia and hedonia beliefs and principles to reduce the negative emotions brought by the conflicts and alleviate the resulting adverse effects.

### 4.4. Contribution and Limitations

The present study on the relationship and mechanisms between academic-athletic conflict and college athletes’ life satisfaction will benefit the academic and practical fields in correctly understanding and addressing the relationship between conflict and well-being. First, according to our model of moderating mediating effects, it is clear that the negative impact of academic-athletic conflict on well-being is not inevitable and that some relevant motivational and emotional variables affect the relationship between the two. This finding explains the variability of the relationship between conflict and well-being across studies. Second, and more importantly, the study verified the role of life motivation in regulating emotions and found differences between eudaimonic and hedonic motives in regulating negative emotions. These findings add details about the regulation of emotions by the two types of motivation and further deepen researchers’ understanding of the relationship between motivation and emotion regulation. Finally, this study provides a reference for the quality-of-life enhancement and motivation development in practical educational contexts targeting the college athlete population. We can suggest and guide students to effectively cope with negative emotions brought about by academics–athletics conflicts and reduce the negative impact of conflicts on well-being experiences by developing good motivation for life.

Although this study on the relationship and mechanism between academics–athletics conflict and college athletes’ life satisfaction greatly benefits to the theoretical and practical fields in correctly understanding and addressing the relationship between conflict and well-being, it still has some limitations. First, according to the bipolar theory of emotions, positive and negative emotions are the two ends of a continuum. An increase in negative emotions is necessarily accompanied by a decrease in positive emotions [42]. Therefore, we examined only negative emotions as the mediating variable to make the model more concise. In contrast, from the perspective of the bivariate model of emotions, an increase in negative emotions may not always predict a decrease in positive emotions [43]. Future attempts will be considered to validate and expand the current study by incorporating both emotions from a bivariate modeling perspective. Second, this study focuses on the moderating effects of eudaimonic and hedonic motives. Studies on emotion regulation suggest that many individual factors regulate negative emotions. Emotion regulation strategies and self-efficacy can also affect the generation and change of negative emotions [38,39]. Future research can conduct a series of experiments and explorations to enrich further the research on the relationship between goal conflict and happiness.

## 5. Conclusions

In order to explore the underlying mechanism of academics–athletics conflict affecting college athletes’ life satisfaction and the effective measures to alleviate it, this study focused on the moderated mediating effect of negative emotions and life motivation between them. The results showed that (1) Negative emotions played an important mediating role between academics–athletics conflict and college athletes’ life satisfaction, with more than 79% of the effects of academics–athletics conflict being achieved through negative emotions. (2) Eudaimonic motives significantly moderated the first half of the mediation path of negative emotions between academics–athletics conflict and life satisfaction. Individuals with high eudaimonic motives experienced fewer negative emotions in the medium and weak conflict conditions. (3) Hedonic motives had a significant moderating effect on the second half of the mediation path. Individuals with high hedonic motives had higher life satisfaction across negative emotion conditions. This study provides important insights for a comprehensive understanding and in-depth study of the relationship between conflict and the well-being of college athletes, as well as a reference for the quality-of-life enhancement and motivation development for college athletes.

## Figures and Tables

**Figure 1 behavsci-13-00093-f001:**
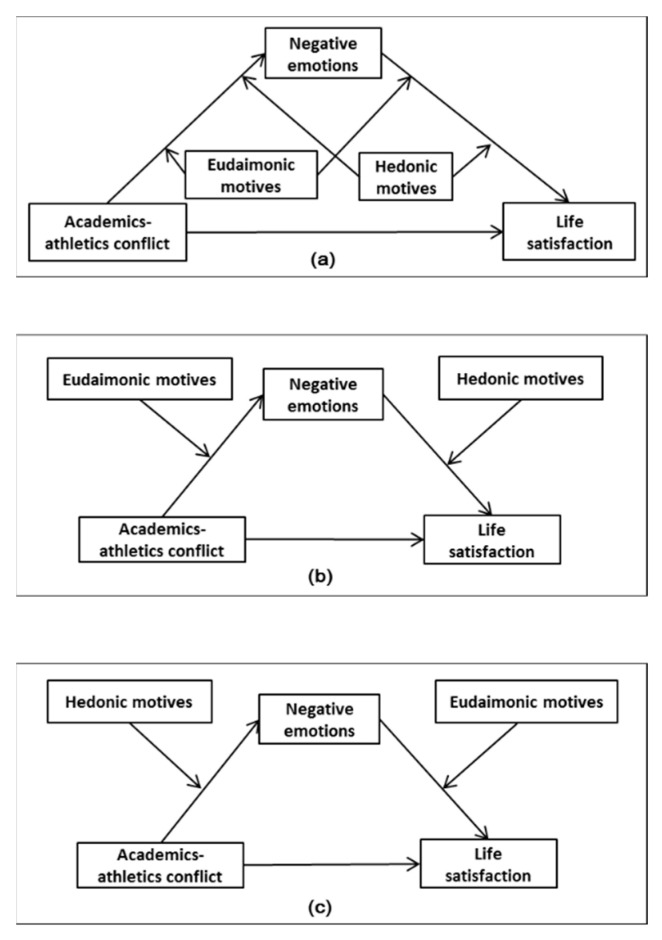
Theoretical model of moderated mediating effect. Subfigure (**a**), (**b**) and (**c**) showed the model proposed in hypothesis 3(a), 3(b) and 3(c), respectively.

**Figure 2 behavsci-13-00093-f002:**
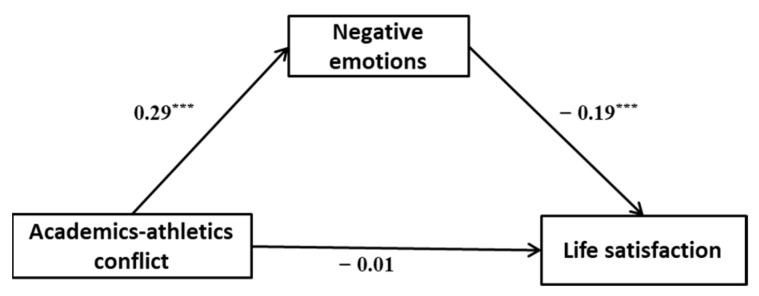
Path coefficients of academics–athletics conflict, negative emotions, and life satisfaction. *** *p* < 0.001.

**Figure 3 behavsci-13-00093-f003:**
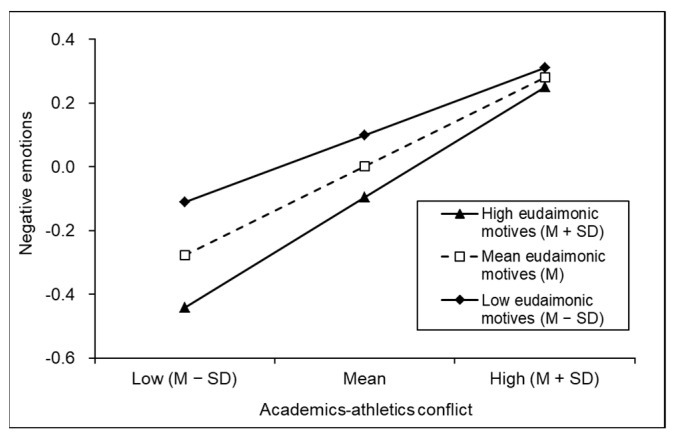
Simple slope diagram of the relationship between academics–athletics conflict and negative emotions mediated by eudaimonic motives.

**Figure 4 behavsci-13-00093-f004:**
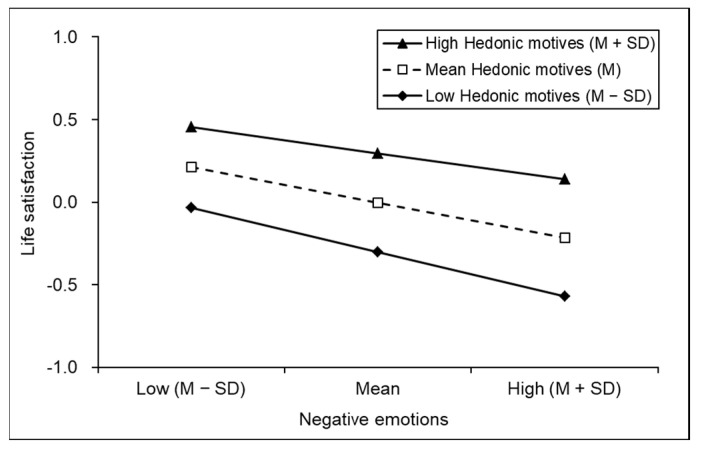
Simple slope diagram of the relationship between negative emotions and Life satisfaction mediated by hedonic motives.

**Table 1 behavsci-13-00093-t001:** Descriptive statistics and correlation matrix of all variables (*n* = 802).

	*M*	*SD*	1	2	3	4	5	6	7	8	9
1. Age	20.01	1.44	1								
2. Gender	-	-	−0.04	1							
3. Grade	-	-	0.47 ***	−0.06	1						
4. Major	-	-	−0.01	−0.11 **	−0.07	1					
5. Academics–athletics conflict	2.98	0.86	0.06	0.03	0.06	−0.03	1				
6. Negative emotions	2.12	0.57	0.03	0.01	−0.02	−0.03	0.29 ***	1			
7. Eudaimonic motives	5.07	1.22	−0.04	0.04	−0.08 *	−0.03	−0.03	−0.12 ***	1		
8. Hedonic motives	4.59	1.23	0.05	0.13 ***	−0.01	−0.01	0.13 ***	0.06	0.53 ***	1	
9. Life satisfaction	4.16	1.35	−0.03	−0.10 **	−0.04	−0.01	−0.07 *	−0.19 ***	0.39 ***	0.27 ***	1

* *p* < 0.05, ** *p* < 0.01, *** *p* < 0.001.

**Table 2 behavsci-13-00093-t002:** Test results of mediating effect of negative emotions.

Regression Equation		Overall Model Fit			Significance of Regression Coefficient			
Outcome	Predictor	*R*	*R* ^2^	*F*	*β*	LLCI	ULCI	*t*
Negative emotions	Gender	0.29	0.83	36.18 ***	−0.01	−0.14	0.12	−0.15
	Academics–athletics conflict				0.29	0.22	0.35	8.51 ***
Life satisfaction	Gender	0.21	0.05	12.55 ***	−0.19	−0.33	−0.05	−2.73 **
	Academics–athletics conflict				−0.01	−0.08	0.06	−0.37
	Negative emotions				−0.19	−0.26	−0.11	−5.13 ***

** *p* < 0.01, *** *p* < 0.001.

**Table 3 behavsci-13-00093-t003:** Regression analysis results of moderated mediating effect.

Regression Equation		Overall Model Fit			Significance of Regression Coefficient			
Outcome	Predictor	*R*	*R* ^2^	*F*	*β*	LLCI	ULCI	*t*
Negative emotions	Gender	0.32	0.10	22.45 ***	0.01	−0.13	0.13	−0.01
	Academics–athletics conflict				0.28	0.21	0.34	8.26 ***
	Eudaimonic motives				−0.10	−0.16	−0.03	−2.88 **
	Academics–athletics conflict × Eudaimonic motives				0.07	0.01	0.13	2.28 *
Life satisfaction	Gender	0.37	0.14	25.61 ***	−0.26	−0.39	−0.13	−3.87 ***
	Academics–athletics conflict				−0.05	−0.12	0.01	−1.53
	Negative emotions				−0.20	−0.27	−0.13	−5.76 ***
	Hedonic motives				0.30	0.24	0.37	9.11 ***
	Negative emotions × Hedonic motives				0.06	0.01	0.11	2.14 *

^*^*p* < 0.05, ^**^
*p* < 0.01, ^***^
*p* < 0.001.

**Table 4 behavsci-13-00093-t004:** The results of moderated mediating effect.

Mediating Variables	Moderating Variables		Effect Size	Boot SE	95% CI	
	Eudaimonic Motives	Hedonic Motives			LLCI	ULCI
Negative emotions	−1.00	−1.00	−0.054	0.020	−0.101	−0.022
	−1.00	0.00	−0.042	0.015	−0.078	−0.017
	−1.00	1.00	−0.030	0.017	−0.074	−0.007
	0.00	−1.00	−0.072	0.019	−0.115	−0.038
	0.00	0.00	−0.055	0.013	−0.083	−0.033
	0.00	1.00	−0.039	0.0176	−0.076	−0.012
	1.00	−1.00	−0.089	0.025	−0.148	−0.047
	1.00	0.00	−0.069	0.015	−0.102	−0.042
	1.00	1.00	−0.049	0.019	−0.088	−0.015

## Data Availability

The data that support the findings of this study are available from the corresponding author upon reasonable request.

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
