# Peer review of "Academics–Athletics Conflict and College Athletes’ Well-Being: The Mediating Effect of Negative Emotions and the Moderating Effect of Life Motivation"

_behavsci, 2023, doi:10.3390/bs13020093_

Round 1

Reviewer 1 Report

The article fully deserves to be published. There is a widespread belief that a scientific career and time devoted to science prevent a professional sports career. As the authors show, the problem lies in the correct resolution of the described conflict. The authors of the article raise a topic that is rarely discussed in the literature. The issue of motivation in experiencing and resolving conflicts should be appreciated not only in the theoretical but also in the practical area.

Author Response

Thanks, please see the revised manuscript for details of other revisions.

Reviewer 2 Report

I have not any question to improve

Author Response

(The authors gave the same response as above.)

Reviewer 3 Report

Dear authors,

The topic of the paper is very interesting and it was my pleasure to read it.

The title of the paper is adequate to the problem of the study. The methodology of the research fits the studied subject. Professional terminology is properly used. As a reviewer of this paper, I find that there are no errors in theoretical presentation, but there is one suggestion to correct in the paper:

At the page 5 line 210 I have notice that authors have stated that "Gender was significantly and positive related to life satisfaction." In the Table 1 the correlation coefficient is "-0.010“ which means that correlation is negative so the correction should be made in the text into "negative correlation".

Parts of the paper are of the appropriate extent and there are no unnecessary repetitions in the text. The text is written clearly and logically, and the conclusion is drowning from the results obtained. As for the literature, references are written correctly.

Sincerely,

Reviewer

Author Response

Thank you for your suggestion, which has been revised in the manuscript. Please see the revised manuscript for details of other revisions.
